# SELF-SUPERVISED DEEP VISUAL STEREO ODOMETRY WITH 3D-GEOMETRIC CONSTRAINTS

## ABSTRACT

This work presents a simple but highly effective self-supervised learning framework for deep visual odometry on stereo cameras. Recent work on deep visual odometry is often based on monocular vision. A common approach is to use two separate neural networks, which use raw images for depth and ego-motion prediction. This paper proposes an alternative approach that argues against separate prediction of depth and ego-motion and emphasizes the advantages of optical flow and stereo cameras. The framework's structure is justified based on mathematical equations for image coordinate transformations. Its central component is a deep neural network for optical flow predictions, from which depth and ego-motion can be derived. The main contribution of this work is a 3D-geometric constraint, which enforces a realistic structure of the scene over consecutive frames and models static and moving objects. It ensures that the neural network has to predict the optical flow as it would occur in the real world.

The presented framework is developed and tested on the KITTI dataset. It achieves state-of-the-art results, outperforming most algorithms for deep visual odometry.

## 1 INTRODUCTION

The *anonymous* project aims to develop a low-cost monitoring system for hard-to-access habitats in the Baltic Sea, using autonomous underwater vehicles. Underwater environments are challenging (Teixeira et al., 2020) due to turbid water and the absence of supporting systems like GPS and most communication systems based on electromagnetic waves. Several requirements for an autonomous underwater vehicle could be formulated based on the experience gained from various experiments within the *anonymous* project. Even though the following requirements are defined for natural underwater habitats, they will most likely apply to many other natural habitats.

Table 1: Requirements for natural habitat monitoring, based on experience in natural underwater environments.

| Requirement | Description |
| --- | --- |
| Precise Odometry | Precise odometry is required for GPS-denied areas. |
| Dense depth | A dense depth estimation is needed to create a complete map of the environment. |
| Robustness | Robustness in challenging conditions is required for use with low light and turbid water. |
| Extendability | To be used in different habitats, the method should be easily extendable without time-consuming data labeling processes. |
| Low cost | Low costs increase the scalability of a system. |

Vision-based systems are very promising for natural (underwater) habitat mapping because they can solve all required tasks while being relatively cheap.

Over the past decades, there has been a lot of research concerning visual odometry and image depth prediction. The classical geometry-based approaches reach high accuracy. However, the robustness of these methods cannot meet the requirements of robots in challenging conditions(Wang et al., 2022).

To solve the problem of robustness, the research field of deep visual odometry uses deep neural networks to predict image depth and ego-motion. Even though these methods do not reach the performance of classical geometry-based methods, they are less sensitive to textureless areas or dynamic environments(Wang et al., 2022). Also, they deliver dense depth predictions as well as ego-motion.

The work of Teixeira et al. (2019; 2020) compares different visual odometry algorithms in an underwater scenario. Their experiments show that classical approaches have great difficulties. In comparison, methods based on deep neural networks have proven to be significantly more robust.

## 1.1 CONTRIBUTION

This paper introduces a simple but highly effective self-supervised learning framework for stereo deep visual odometry. The framework is based on a deep neural network for optical flow predictions, from which depth, ego-motion, and moving objects can be detected. The main contribution of this work is a 3D-geometric constraint, which enforces a realistic structure of the scene over consecutive frames and models static as well as moving objects.

Since deep visual odometry is a general research field of robotics, and no qualitative dataset of the Baltic Sea is available, the presented framework is developed and tested on the KITTI dataset. It achieves state-of-the-art results, outperforming most algorithms for deep visual odometry.

## 2 RELATED WORK

While classical geometry-based visual odometry methods have proven to be very accurate under favorable situations, they often fail under challenging conditions like textureless areas or dynamic environments(Wang et al., 2022). The emerging field of deep visual odometry aims to solve these problems using deep neural networks. The work of Wang et al. (2022) gives a good overview of the current development in deep visual odometry.

Large amounts of data are needed for supervised training of deep neural networks. For this purpose, datasets like KITTI (Geiger et al., 2012; Menze et al., 2015; 2018) and others were created. Unfortunately, creating labeled datasets for neural network training is very time-consuming and expensive. To solve this problem, Zhou et al. (2017) proposed a method in which deep visual odometry can be learned in a self-supervised fashion by minimizing the photometric differences between a synthesized and an original target view.

However, the problem statement of photometric consistency is ill-posed because light conditions can differ from frame to frame. To solve this problem Zhan et al. (2018) uses features created by deep neural networks instead of pixels.

Many researchers try to solve the task of deep visual odometry by working directly on images. A common approach used by Zhou et al. (2017); Mahjourian et al. (2018); Yin & Shi (2018); Zhan et al. (2018); Ranjan et al. (2019) is to use two different convolutional neural networks. One neural network is used to predict the depth of an image (with monocular vision), and the other neural network is used to predict the ego-motion from two or more consecutive frames.

The image space is generally very difficult to work with because it has many complex and redundant information. Several researchers (Costante & Ciarfuglia, 2018; Zhao et al., 2021; 2022) use optical flow as features for pose estimation to simplify the task.

To increase the accuracy of deep visual odometry, several researchers use geometric constraints: Mahjourian et al. (2018) create point clouds from the depth map and camera intrinsics, enforcing their consistency to the movement over consecutive frames. Zhao et al. (2022) use the optical flow to enforce depth consistency over consecutive frames.

The training of deep visual odometry and the constraints used to improve the quality often rely on the assumption of a static world. This assumption is violated by moving objects, which results in

the degradation of training quality. Ranjan et al. (2019) tries to solve the moving object problem with a competitive game where a visual odometry network tries to explain the world using a static world model, movement, and depth, playing against an optical flow model with no constraints. Zhao et al. (2022) tries to detect moving objects and occluded regions by detecting differences between the forward optical flow and the inverse backward optical flow.

From all deep visual odometry methods reviewed in this work, the method of Yang et al. (2018; 2020) shows the highest precision for both visual odometry and depth prediction. Characteristics of their method include training on stereo images, correcting lightning conditions of images, and modeling the photometric uncertainties of pixels.

Most algorithms for deep visual odometry have been developed on datasets for autonomous driving. However, the work of Teixeira et al. (2019; 2020) compares the algorithm developed by Zhou et al. (2017) and Yin & Shi (2018) with classical geometry-based algorithm in an underwater scenario. Their experiments show that classical approaches have great difficulties. In comparison, methods based on deep neural networks have proven to be significantly more robust.

## 2.1 CONCLUSION

The research of deep visual odometry is often based on the work of Zhou et al. (2017). Many researchers (Mahjourian et al., 2018; Yin & Shi, 2018; Zhan et al., 2018; Ranjan et al., 2019; Zhao et al., 2022) adopt their method and use one neural network for monocular depth prediction and another neural network for visual odometry. Both neural networks are trained (mostly) self-supervised using the photometric consistency.
While monocular vision has the advantage of only needing one camera, it is mathematically impossible to compute the depth at the correct scale without a reference like a second camera with known extrinsic parameters. Despite this, our literature review shows that neural networks have become very good at guessing the correct scale (presumable by using the size of known objects). However, this can not be guaranteed in natural environments where no standardized-sized objects exist.

To compute the photometric consistency, the optical flow between two images is computed, which can be done mathematically using image depth and ego-motion.

We argue that image depth and ego-motion should not be computed separately since the optical flow closely connects them. Therefore, the optical flow, which can be learned directly from consecutive frames, should be the basis for image depth and ego-motion prediction. This argument is supported by Costante & Ciarfuglia (2018); Zhao et al. (2021; 2022), who used the optical flow as a feature for visual odometry.

Although the above-mentioned methods show very good solution approaches, they do not reach the performance of classical approaches on benchmarks like KITTI. Therefore, this work tests a different approach derived from 3D-geometric constraints describing the relation between camera perspectives and world coordinates.

## 3 METHOD

This section describes the framework for stereo deep visual odometry. Please note that this section is meant to give an understanding of the methodology and does not provide all the information that is required to reproduce the experiments. For this purpose, the source code is publicly available at: [to be added at acceptance]

### 3.1 MATHEMATICAL ANALYSIS

If we consider a static scene, the optical flow $F_m$ of each pixel position $\mathbf{p} = [u, v, 1]^T$ caused by the ego-motion of a camera is directly connected to the transformation matrix $T$, the image depth $D$ and the camera intrinsic matrix $K$. The connection can be described by equation 1:[1]

---

[1]Note that $D_{t+1}^{u,v}$ can be directly derived from the equation itself. The term $K T_{t \to t+1} D_t^{u,v} K^{-1} \mathbf{p}$ calculates the pixel coordinate in the three-dimensional space in which $D_{t+1}^{u,v}$ is the third element.

$$F_m^{u,v} = \frac{1}{D_{t+1}^{u,v}} K T_{t \to t+1} D_t^{u,v} K^{-1} \mathbf{p} - \mathbf{p} \tag{1}$$

A simplification of this equation is the image Jacobian matrix.

$$\begin{pmatrix} \dot{u} \\ \dot{v} \end{pmatrix} = \begin{pmatrix} -f/d & 0 & u/d & uv/f & -(f+u^2/f) & v \\ 0 & -f/d & v/d & f+v^2/f & -uv/f & -u \end{pmatrix} \begin{pmatrix} v_x \\ v_y \\ v_z \\ \omega_x \\ \omega_y \\ \omega_z \end{pmatrix} \tag{2}$$

The Jacobian matrix describes the pixel position derivative $(\dot{u}, \dot{v})$ (which is approximates the optical flow), depending on the pixel position $(u, v)$, the camera focal length $f$, the image depth $d$ and the ego-motion of the camera $(v_x, v_y, v_z, \omega_x, \omega_y, \omega_z)^T$. Since it is easier to understand, the image Jacobian matrix is used to argue about the mathematical background of visual odometry (the references to the equation are outsourced to the footnotes). However, in the implementation, the equation 1 is used.

For this work, we consider a completely visual-based odometry approach, where only the camera information, meaning the focal length $f$ and the pixel positions $(u, v)$ are known. In this case, the optical flow, the image depth, and the ego-motion, which consists of a translational and a rotational component, is unknown.

The optical flow can be predicted, independent of all other parameters, by finding corresponding patterns between two images. Successful methods can be found in the rankings of different optical flow datasets like KITTI (Menze et al., 2015; 2018) and others. Currently, methods based on neural networks show the best performance. The training of optical flow networks can be conducted without any labeled data using photometric consistency.

There are three common ways to estimate the image depth. One is to learn the depth of a single image from experience by training on large amounts of labeled data. This method is unsuitable for our purposes because it cannot be easily extended to new environments due to the need for labeled data.

Another method is to learn the depth from motion using consecutive frames of a single camera. This can be done without the need for labeled data. However, estimating the depth from motion only works with translational movements since the optical flow caused by rotation is independent of the depth [2]. Also, this method is affected by the problem of scalability, meaning that the depth and translational movement can only be estimated up to a scale factor [3]. To know the true scale, some reference measure is needed. Another disadvantage is that most methods, which are reviewed in this work, rely on a static world assumption. This assumption can be violated by moving objects, which results in the degradation of training quality.

The simplest way to estimate the depth is by using a stereo camera. The baseline distance (in x or y direction) between the two cameras causes a homogeneous optical flow where change arises only through the depth[4]. Thereby, the depth can be directly calculated with the optical flow and the stereo camera baseline. Another advantage is that two cameras can be triggered at the same time. Without a time difference, the scene is completely static, which simplifies the situation since all image movements can only be caused by the stereo camera baseline. For the above-mentioned reasons, We argue that the advantages of adding a second camera to a given setup can outweigh the additional costs in many cases.

To our knowledge, the ego-motion can only be estimated from images by observing optical flow between two or more consecutive frames. While the rotation can be estimated correctly from the optical flow, the translational movement can only be estimated up to a scale factor unless the correct image depth is known.

---

[2] The optical flow computation in equation 2 shows that the depth $d$ only effects the optical flow caused by the translational movements $(v_x, v_y, v_z)^T$.

[3] When scaling the translational movements $(v_x, v_y, v_z)^T$ and depth $d$ in equation 2 with the same scale factor, the scale factor cancels itself out.

[4] With only one non zero movement, for example in x direction $v_x$, equation 2 can be reduced to $\dot{u} = -f/d$, where $d$ is the only unknown variable

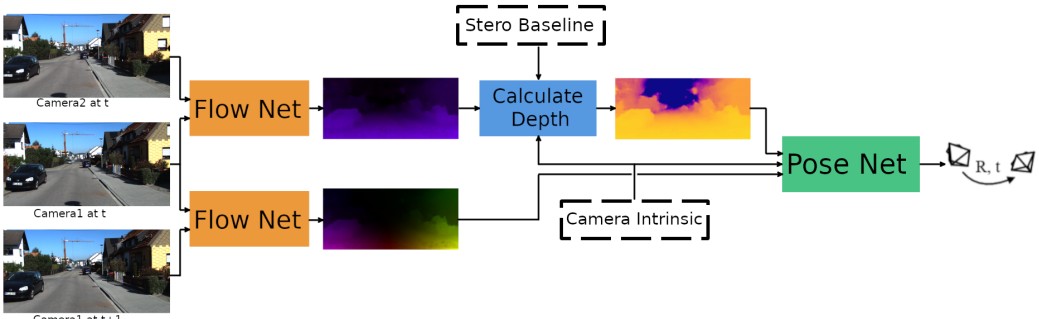

Figure 1: Framework for deep visual stereo odometry.

## 3.2 SOLUTION APPROACH

Based on the analysis of equation 1 and 2, this work uses optical flow detection for depth and ego-motion estimation. Also, an adequately rectified stereo camera is used for the above reasons. The entire setup of the visual odometry can be seen in figure 1.

In the first step, the optical flow between the two cameras is predicted using a neural network referred to as FlowNet. The image depth can be directly calculated using the optical flow, the camera intrinsic matrix, and the stereo baseline. For a well-rectified stereo camera setup, the offset between both cameras results in only one translational movement (e.g. in x direction). Therefore, the equation 2 can be simplified and rearranged to [5]:

$$d = -\frac{f}{\dot{u}} * v_x \tag{3}$$

where $v_x$ is the known baseline distance between the two cameras.

The ego-motion can be calculated based on the optical flow between two consecutive frames of one camera and the image depth. Since the ego-motion consists of six components, knowing the optical flow, depth, and camera intrinsic matrix but not the ego-motion still leaves equation 2 to be an underdetermined system of equations. For this reason, multiple points are needed to estimate the correct ego-motion. One option is to use classical geometry-based methods like the eight-point algorithm. However, these methods are not robust against inaccurate flow predictions. To ensure robustness, this work uses a deep neural network for ego-motion prediction. This neural network, referred to as PoseNet, uses the optical flow, image depth, and camera intrinsic information as input features to solve the problem of pose prediction.

## 3.3 LOSS FUNCTIONS

The main loss function of the FlowNet and the PoseNet is the photometric consistency loss, which is computed between a target images $I_t$ and a source image $I_s$ which is warped to the target coordinate frame using the optical flow.

$$L_{pho} = \sum_{u,v} ||I_t^{u,v} - warp(I_s^{u,v}, flow)|| \tag{4}$$

One of the main contributions of this work is the 3D geometric constraint.[6] Using the predicted depth $D$ and the camera intrinsic matrix $K$, each pixel can be transformed into a point in 3D space $Q^{u,v} = D^{u,v}K^{-1}[u,v,1]^T$. Considering the point cloud $Q_t$, $Q_{t+1}$ and the ego-motion $T_{t \rightarrow t+1}$ in

---

[5]Simplifying equation 1 with a camera motion of $(v_x, 0, 0, 0, 0, 0)$, yields the same results

[6]The 3D geometric constraint is inspired by Mahjourian et al. (2018).

a static world, the following equation would apply: $Q_t = T_{t \to t+1} Q_{t+1}$ [7]. In a dynamic world, the difference between the point clouds $\Delta Q = Q_t - T_{t \to t+1} Q_{t+1}$ should correspond to the movement of a self-moving object. Therefore, the optical flow created by the movement of $\Delta Q$ should be the same as the difference between the optical flow and the optical flow caused by the ego-motion. The above description is formulated as a loss function, described in the algorithm 1. One difficulty is connecting the point cloud's corresponding points, which is solved by using the optical flow.[8]

---

**Algorithm 1** 3D geometric constraint

---

**Require:** $f, f_{move}, D_{t+1}, T_{t \to t+1}, K$ **return** $L_{cons3D}$
   $Q_{t+1} = D_{t+1} K^{-1} p$
   $\hat{Q}_t = T_{t \to t+1} Q_{t+1}$
   $\overline{Q}_t = warp(\hat{Q}_t, f)$
   $f_{add} = p - K \overline{Q}_t$
   $L_{cons3D} = ||f_{add} - f - f_{move}||$

---

## 4    EXPERIMENTS

For evaluation of the proposed framework, deep neural networks are trained and tested on the KITTI dataset. To get high-quality evaluation data, the results for the test data of the KITTI odometry, stereo, optical flow, and scene-flow[9] datasets are uploaded to the official KITTI evaluation server. To better compare the depth prediction with related methods, our method is also validated on the KITTI Eigen split (Eigen et al., 2014).[10] The following sections describe the neural networks and the training process and compare the odometry and flow data results with state-of-the-art methods. Since none of the related work was evaluated on the KITTI stereo or scene-flow dataset, these results are not further discussed. However, the reader is welcome to look at the official results and compare them with other methods. The results can be found on the KITTI website[11] under the name 3DG-DVO (3D-Geometric constrained Deep Visual Odometry).

### 4.1    NEURAL NETWORKS

**Fully convolutional variational autoencoder:** The photometric consistency is not computed directly on the image but on the latent space of a fully convolutional variational autoencoder. The autoencoder is trained using the image reconstruction and the Kullback-Leibler-Divergence. Applying the photometric consistency to an embedding results in a more stable and faster training. Also, it has the potential to facilitate training because, in contrast to raw pixels, which suffer from changing light conditions, the features of the autoencoder can be trained to be consistent between frames.

**FlowNet:** In theory, every kind of optical flow predictor, which creates a dense optical flow, can be used as the FlowNet. This work uses a customization of the PWC-Net (Sun et al., 2018), since it has high performance while being sufficiently efficient.

**PoseNet:** The PoseNet used in this work consists of the following sequential layers: a *flipping layer*, four sequential convolutional layers, a global average pooling layer, and a dense layer for ego-motion prediction. A flipping layer is a custom layer that flips the input features around the vertical, horizontal, and diagonally axis and concatenates the flipped features at the channel axis. This layer is inspired by the eight-point algorithms and extends the reception field of the PoseNet. By flipping the input feature, one convolutional filter can compare the optical flow from distant corners of the image, facilitating the detection of movements that create symmetric flow fields around the camera's focal point.

---

[7]The point clouds have the opposite movements as the ego-motion.

[8]Using the optical flow to connect corresponding points is inspired by the depth consistency loss of Zhao et al. (2022)

[9]The KITTI scene-flow dataset is a combination of stereo depth and optical flow prediction

[10]For depth prediction, the official test data can not be used because it is specifically meant for mono depth prediction.

[11]https://www.cvlibs.net/datasets/kitti/

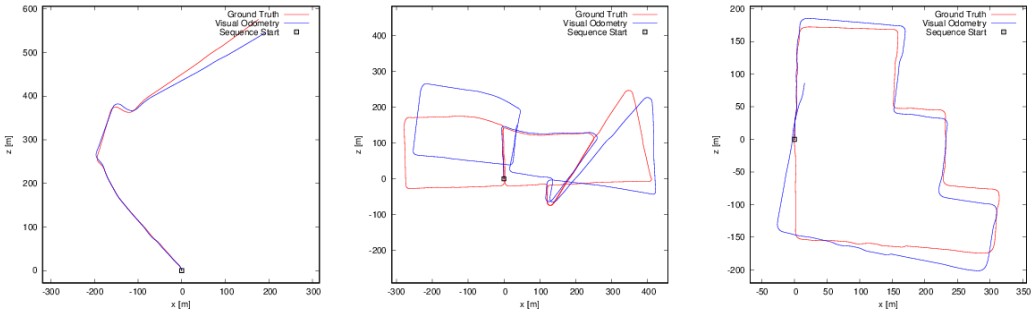

Figure 2: Trajectories on the KITTI odometry test dataset.

## 4.2 TRAINING

The training is conducted on the official training data (sequence 0-10) of the KITTI odometry dataset.

For stability purposes, at the beginning, only the autoencoder is trained, successively joined by the FlowNet and the PoseNet. The 3D geometric constraints and the gradient flow between FlowNet and PoseNet are neglected at the beginning of the training and then linear increased. For the main training process, the gradient flow of all neural networks is connected.

## 4.3 EVALUATION OF VISUAL ODOMETRY

Table 2 shows the performance of the KITTI odometry test dataset. Only methods are compared which uploaded their data to the official validation server. Images of the trajectories can be seen in figure 2.

Table 2: Comparison on the KITTI odometry test dataset.

| Method | Translation Error [%] | Rotation Error [deg/m] |
|---|---|---|
| Bian et al. (2019) | 21.47 | 0.0425 |
| Ranjan et al. (2019) | 16.06 | 0.0320 |
| Godard et al. (2019) | 12.59 | 0.0312 |
| Zou et al. (2020) | 7.40 | 0.0142 |
| Yang et al. (2020) | 0.88 | 0.0021 |
| Ours | 6.77 | 0.0125 |

The results show that our method achieves state-of-the-art performance in visual odometry. Only the algorithms of Yang et al. (2020) achieves better results.

## 4.4 EVALUATING OPTICAL FLOW

The official optical flow evaluation results are compared with the work of Sun et al. (2018) and Ranjan et al. (2019) (see Table 3). Sun et al. (2018) developed the here used PWC-Net but used supervised training. Ranjan et al. (2019) is the only related work of deep visual odometry that submitted optical flow test data to the official KITTI evaluation server.

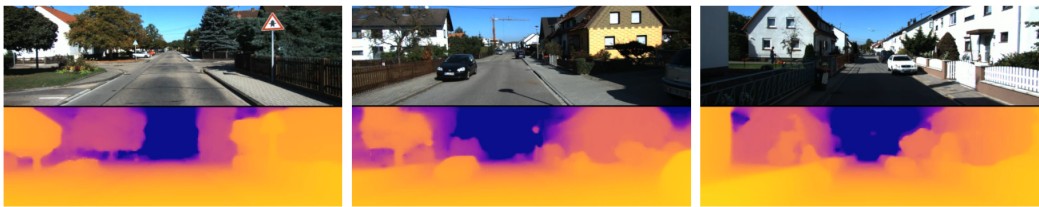

Figure 3: Depth prediction images.

Table 3: Comparison of the percentage of optical flow outliers (Fl). Pixels are considered correctly if the disparity or flow end-point error is $< 3px$ or $< 5\%$. The results are only for non occluded regions (Ranjan et al. (2019) does not specify the evaluation region).

| Method | Fl |
|---|---|
| Sun et al. (2018) | 6.12 % |
| Ranjan et al. (2019) | 25.27 % |
| Ours | 28.68 % |

The comparison shows that the flow prediction of our method is slightly worse than the work of Ranjan et al. (2019) and significantly worse than the work of Sun et al. (2018), which used the same neural network architecture with supervised training.

## 4.5 EVALUATING DEPTH PREDICTION

The depth predictions are validated on the KITTI Eigen split (Eigen et al., 2014) and compared with related deep visual odometry methods (see Table 4). Images of the depth predictions can be seen in figure 3. Please note that one of the main differences between our work and related deep visual odometry methods is the use of a stereo camera, which gives it an advantage over monocular camera-based methods. On the one hand, it makes the comparison to related methods difficult, but on the other hand, it underlines the benefits of a stereo camera, which is one of the key arguments of this work.

Table 4: Comparison of the depth prediction on the KITTI Eigen split. Please note that one of the main differences between our work and related deep visual odometry methods is the use of a stereo camera, which gives it an advantage over monocular camera-based methods.

| Method | Error Metric | | | | Accuracy Metric | | |
|---|---|---|---|---|---|---|---|
| | Abs.Rel | Sq.Rel | RMSE | RMSE (log) | $\delta < 1.25$ | $\delta < 1.25^2$ | $\delta < 1.25^3$ |
| Zhou et al. (2017) | 0.198 | 1.836 | 6.565 | 0.275 | 0.718 | 0.901 | 0.960 |
| Mahjourian et al. (2018) | 0.159 | 1.231 | 5.912 | 0.243 | 0.784 | 0.923 | 0.970 |
| Yin & Shi (2018) | 0.153 | 1.328 | 5.737 | 0.232 | 0.802 | 0.934 | 0.972 |
| Ranjan et al. (2019) | 0.139 | 1.032 | 5.199 | 0.213 | 0.827 | 0.943 | 0.977 |
| Zhao et al. (2022) | 0.136 | 1.031 | 5.186 | 0.209 | 0.831 | 0.947 | 0.981 |
| Yang et al. (2020) | 0.099 | 0.763 | 4.485 | 0.185 | 0.885 | 0.958 | 0.979 |
| Ours | 0.046 | 0.233 | 2.455 | 0.088 | 0.978 | 0.993 | 0.996 |

The results in Table 4 show that the depth prediction of my method outperforms all deep visual odometry methods that are reviewed in this work.

## 5 CONCLUSION AND FURTHER RESEARCH

This work introduces a simple but highly effective self-supervised learning framework for deep visual odometry on stereo cameras. The experiments on the KITTI dataset show that the framework reaches state-of-the-art performance for visual odometry and optical flow predictions. For depth predictions, it outperforms all monocular camera-based methods.

Based on the experiment results, we want to underline the benefits of stereo cameras, which are most likely the reason for the outstanding results in depth prediction. With the use of a stereo camera, nearly the entire problem of deep visual odometry can be solved by one neural network for optical flow prediction, which is guided by a model for realistic movements. I emphasize that current research in deep visual odometry should take advantage of hardware that is adapted to the task and concentrate on problems like robust feature matching that cannot be sufficiently solved with hardware or classical methods.

In further research, the proposed framework for stereo deep visual odometry will be tested in different natural environments.
For one anonymous project, the framework will be tested for its usability in natural underwater habitats of the Baltic Sea.
In another anonymous project, it will be used to create a digital twin of apple farms.

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
