# OpenReview forum: "Self-Supervised Deep Visual Stereo Odometry with 3D-Geometric Constraints"
_ICLR.cc/2024/Conference — ICLR 2024 Conference Withdrawn Submission_

### Official Review · Reviewer_TCz4 · 2023-10-27

**Soundness:** 3 good
**Presentation:** 4 excellent
**Contribution:** 2 fair
**Rating:** 6
**Confidence:** 4

**Summary:**

The paper introduces a self-supervised deep-learning framework for visual odometry and depth reconstruction using stereo cameras. The proposed framework is designed to meet the requirements of a monitoring system for underwater environment and other natural habitats in which classical geometric computer vision methods have low robustness.

The main contribution of the paper is to demonstrate the benefits of utilizing stereo cameras for this task, in particular for depth reconstruction. This outcome is apparent when results on Tables 2 and 3 are contrasted with results on Table 4. Results on Table 2 demonstrate that the proposed method is not the most accurate for the prediction of camera motion. Moreover, results in Table 3 demonstrate that the proposed method is not the most robust for estimation of optical flow. However, despite these disadvantages the proposed method is clearly the most accurate at depth prediction, as can be seen on Table 4.

An additional contribution of the paper is the joint estimation of optical flow across the cameras in the stereo pair, for which an assumption of a static scene is always true for synchronized cameras, and across frames, for which such assumption is no longer valid. This network takes as inputs the intermediate representation of a variational autoencoder as input.

**Strengths:**

# Originality
The main claimed contribution of the paper is a demonstration of the clear benefits of utilizing stereo rather than monocular systems for depth reconstruction. In the paper, these benefits can even compensate for lower-quality inputs. This conclusion is supported by the superior accuracy in depth reconstruction in comparison to monocular methods even when the latter produce superior optical-flow inputs for depth estimation or camera-pose outputs. This is not a trivial result; in fact, the paper supports and could equally claim the negative result that stereo does not bring immediate benefits to the related problems of estimating optical flow and camera pose.

# Quality
The paper is well situated among its peers and brings a good review of the relevant literature. The conclusions presented in the paper are justified and not overblown. The experimental section of the paper is convincing insofar as it applies to traditional datasets such as KITTI.

# Clarity
The paper is exceedingly well written. It is largely self-contained, avoiding unnecessary jargon. Its language is clear and does not overcomplicate simple concepts. The switching between equations 1 and 2 for implementation and exposition, respectively, is an interesting and effective pedagogical device. The imbalance in the comparison of the proposed stereo-based method with monocular systems is readily and clearly acknowledged.

# Significance
The paper adds to the growing body of knowledge that combines classical computer-vision methods with deep learning. It does so by, in a straightforward and clear manner, demonstrating how classical methods (the depth-reconstruction component of the proposed framework) yield good results when provided with features (broadly defined here as the optical-flow component of the framework) estimated through a deep learning method. It also, and perhaps inadvertently, reaffirms the notion (right or wrong) that classical methods are typically more accurate but less robust than those based on deep learning for regression problems, such as camera-pose estimation.

**Weaknesses:**

What is most surprising about the paper is that not *everything* is improved by the use of a stereo camera pair when compared to a monocular system - the results in the paper show a clearly disadvantage of the stereo-based method for estimation of optical flow and camera pose. A major weakness of the paper is therefore that it does not even propose a hypothesis for the cause of this unexpected result. As it stands, this provocative issue was graciously acknowledged but no further discussed.

It is understandable that validation of the method and comparison against other algorithms for the underwater monitoring scenario which motivates the work may not be practical, due to the lack of standardized datasets. However, there should be some justification as to if and why, given the shortcomings in the results in comparison to other methods, particularly in the robustness of the estimated optical flow, the proposed framework would indeed generalize to scenarios radically different from those covered by the KITTI dataset.

The names "FlowNet" and "PoseNet" proposed in section 4.1 are "taken," have already been applied to similar contexts in https://arxiv.org/abs/1612.01925 and https://arxiv.org/abs/1509.05909, respectively. In the same section, it is not clear what in the eight-point algorithm is connected to the flipping layer it purportedly inspired.

**Questions:**

# General
Why of all intrinsic parameters only focal length is assumed to be known, as stated in section 3.1? Why not assume that all intrinsics have been fully calibrated? It seems this would require only a small additional effort to that of estimating the focal length.

The paper is well positioned to propose an explanation as to how the estimation of optical flow using a stereo system could produce inferior results to those of a monocular one, at least measured in percentage of outliers. This is an important issue, because the general claim (not only of the paper, but in much of the literature) is that deep-learning methods offer greater robustness. How does this advantage play for stereo systems? Even though the comparison of the estimated optical flow in the paper is not against classical methods (which in the current state of the art are known to be inferior to methods utilizing deep learning), any insight into this question is relevant. More specifically: Why doesn't the customized PWC-Net used in the paper yield results superior to that of Ranjan et al. (2019), as shown in Table 3?

A related question is what is special about the problem of depth estimation, if anything, that allows it to benefit from the availability of stereo image pairs, while the computation of optical flow, according to the paper, does not? Since optical flow is scale free while depth isn't, the hypothesis that the calibrated baseline of a stereo camera pair is the relevant factor, for example, could be proposed. A fuller analysis of this issue is well within the scope of the paper.

How does the simplification of equation 2 behave in the presence of other intrinsic parameters, such as the principal point and lens distortion?

It is unclear what benefit is added by the use of the fully convolutional variational autoencoder to generate inputs to the network that computes optical flow. It is actually not entirely clear that this is what happens, as the variational autoencoder is missing in Figure 1. But if it is, the backbone of PWC-Net if Sun et al., on which the paper's "FlowNet" is based, is a U-Net, which should perform a similar task to that of the variational autoencoder. Why isn't that sufficient?

# Typos
## Significant
- Multiple places: vertical space missing between
    - paragraphs 1 and 2 of the Abstract
    - paragraphs 2 and 3 of section 1
    - paragraphs 1 and 2 of section 1.1
    - paragraphs 2 and 3, and 4 and 5 of section 2
    - paragraphs 1 and 2 of section 2.1
    - most paragraphs on page 4
    - paragraphs 1 and 2 of section 4
    - paragraphs 1 and 2 of section 4.2
    - paragraphs 3 and 4 of section 5
- On page 2, paragraphs 1, 2, and 5:  A space is missing between the parenthesis introducing the reference to "Wang et al., 2022" and the word that precedes it.
- On page 4: "the optical flow, the image depth, and the ego-motion [...] are known," instead of "is known."

## Minor
- The footnote to equation 1 appears on the page that precedes it, and therefore that footnote is confusing until we turn the page and come back. Perhaps the footnote could be added to the equation directly.
- On page 4, at the end of the paragraph following equation 1, it may be better to write "However, in the implementation, equation 1 is used."
- There is a jump from the plural "we" used throughout the paper to a singular "I" in the second paragraph of section 5.

---

### Official Review · Reviewer_GRTx · 2023-10-28

**Soundness:** 1 poor
**Presentation:** 1 poor
**Contribution:** 1 poor
**Rating:** 1
**Confidence:** 4

**Summary:**

The authors propose a self-supervised learning framework for deep stereo visual odometry. They use a network to predict dense optical flow to estimate depth and dynamic objects. Then, they utilize posenet to estimate camera pose.

**Strengths:**

The paper is well-written. It introduces a constraint stating that the optical flow generated by the motion of ∆Q should be equal to the difference between the optical flow and the optical flow caused by ego-motion.

**Weaknesses:**

1. Use the incorrect template, should be ICLR 2024 instead of ICLR 2023.
2. An incomplete reference
3. The experimental results are not better than the baseline, and the experiments are not comprehensive. For example, for the comparison of trajectory accuracy, the EuRoC dataset can be used. There is also no comparison with the state-of-the-art in some SLAM methods, such as ORB-SLAM3.

**Questions:**

Pose estimation is well formulated by geometry constraints. Why introduce PoseNet to solve it? It seems to create an explainable question in an unexplainable manner. And also it doesn't improve accuracy.

---

### Official Review · Reviewer_5ug3 · 2023-10-30

**Soundness:** 1 poor
**Presentation:** 1 poor
**Contribution:** 1 poor
**Rating:** 1
**Confidence:** 5

**Summary:**

This paper proposes a self-supervised learning method to address the visual odometry problem, in which the authors emphasise the importance of using stereo cameras and optical flow. However, both of them are not new to this topic. More importantly, it looks like an incomplete submission. The presentation, reference, methodology, and the experiments are insufficient.

**Strengths:**

The proposed method is slightly different from previous methods, even though the use of optical flow and stereo cameras in this field has not been new.

The depth accuracy of the proposed method is slightly better than previous methods that are mentioned in the paper.

**Weaknesses:**

The paper looks incomplete. I hope that the authors can significantly revise the paper and add more content for the next submission.

The novelty of the proposed method is very limited. Please reconsider this problem and better claim the contribution.

The evaluation results show that the proposed method is worse than previous methods in both visual odometry and optical flow estimation.

**Questions:**

NA

---

### Official Review · Reviewer_4Waj · 2023-10-31

**Soundness:** 1 poor
**Presentation:** 2 fair
**Contribution:** 1 poor
**Rating:** 3
**Confidence:** 4

**Summary:**

This paper introduces a self-supervised learning framework for deep visual odometry using stereo cameras. Unlike traditional approaches that use separate neural networks for depth and ego-motion prediction based on monocular vision, this work advocates for a unified approach, emphasizing the benefits of optical flow and a stereo camera setup. The framework relies on a deep neural network for optical flow prediction for deriving depth and ego-motion information. The key contribution is a 3D-geometric constraint that enforces realistic scene structure over consecutive frames, accounting for both static and moving objects. The framework is evaluated on the KITTI dataset and argues state-of-the-art performance.

**Strengths:**

The positive aspects of the paper are:
(1) The paper discusses adequately the difficulties of underwater scenarios for visual odometry computation, enumerating some important requirements for the system to work in natural environments.

**Weaknesses:**

The negative aspects of the paper are:
(1) As highlighted in Section 1.1, the primary contribution is "3D geometric constraint, which enforces a realistic structure of the scene over consecutive frames and models static as well as moving objects." This contribution is discussed in Section 3.3. where it is mentioned that this idea is inspired by Mahjourian et al. (2018). However, it remains somewhat unclear how the proposed Algorithm 1 differs from the approach in Mahjourian et al. (2018). The latter focused on explicitly considering inferred 3D geometry for the entire scene and enforcing consistency in 3D point clouds and ego-motion across consecutive frames, solved through a novel backpropagation algorithm. The distinction between these contributions is not well articulated in the paper, and is essential to understand the relevance of the proposed work.
(2) A significant portion of the mathematical analysis in Section 3.1 could be moved to the supplementary material as it primarily covers standard theoretical content.
(3) Looking at Figure 1 and from Section 4.1., the main building blocks of the pipeline are FlowNet and PoseNet. These blocks are based on prior work without relevant novelty.
(4) In the final sentence of Section 4.3, the authors assert that their method achieves state-of-the-art performance in visual odometry. However, a closer examination of Table 2 reveals that Yang et al. (2020) outperforms the proposed method by a significant margin, raising concerns about the veracity of this statement.
(5) In Table 3, the proposed approach (Ours) is worse than previous works, even when considering that some of these prior works are over four years old.
(6) The proposed approach (Ours) in Table 4 is the top performer, but this is expected, as it is the only work using stereo cameras. Stereo imagery considerably simplifies depth prediction in terms of accuracy, robustness and removes the ill-posed aspect of monocular approaches.

**Questions:**

In order to change my opinion, I would like the authors to comment on some of the points enumerated in the Weaknesses Section:
(1) The authors should provide a clear and detailed explanation of the specific distinctions between their proposed 3D geometric constraint (Algorithm 1) and the approach discussed in Mahjourian et al. (2018). They should elaborate on the novel aspects or improvements in Algorithm 1 compared to the method inspired by Mahjourian et al. (2018). This will help readers understand the unique contribution of the paper.
(2) The authors should comment on the novelty of FlowNet and PoseNet in the context of their work.
(3) The authors claim state-of-the-art performance, but there is a discrepancy of this statement when analyzing Table 2 and Table 3, where their method appears to underperform compared to other works.

General comments:
(1) The authors should specify the hardware on which their pipeline was implemented and provide runtime information. This information is essential for readers to understand the practical feasibility and efficiency of the proposed method.
(2) I don't completely understand the statement in Section 4.1. ".... This layer is inspired by the eight-point algorithms and extends the reception field of the PoseNet." Can you comment on this please.
(3) The authors should consider including references in the first paragraph of Section 4.1 to provide additional context and sources.